# Arbuscular mycorrhizal fungi improve drought toleration in *Cinnamomum migao* H.W.Li seedlings by increasing plant growth, nutrient uptake and biomass accumulation

Xiao Xuefeng[1,2☯], Tian Xiu[2,3☯], Hao Gang[1]*, Xu Lu[4], Huang Rui[5], Zan Yue[1]

**1** Suzhou Polytechnic University, Suzhou, Jiangsu, China, **2** Guizhou University, Guiyang, Guizhou, China, **3** Guizhou Yudi Technology Co., Ltd, Guiyang, Guizhou, China, **4** Jiangsu Agri-animal Husbandry Vocational College, Suzhou, Jiangsu, China, **5** Suzhou Polytechnic Institute of Agriculture Suzhou, Suzhou, Jiangsu, China

☯ These authors contributed equally to this work.
* 626052286@qq.com

## Abstract

Drought stress is a primary factor reducing field crop productivity, and its impact is predicted to intensify and occur more often because of human-influenced environmental and climate changes. Which exerts a critical influence on plant growth and distribution, especially in semi-arid Karst regions including southwest China. *Cinnamomum migao* H.W.Li (*C. migao*), a tree in the Cinnamomum genus of Lauraceae family, is a medicinally important tree species endemic to southwest China. Arbuscular mycorrhizal fungi (AMF) symbiosis mitigates drought stress in plants, yet the inoculation method affects the establishment and function of this symbiosis remains unclear. Therefore, we conducted an experiment examining the influence of different AMF (*Funneliformis mosseae* (*F. mosseae*) and *Claroideoglomus etunicatum* (*C. etunicatum*) their combination (Mixed)) on *C. migao* seedlings. AMF colonization rates, root vigor, seedling growth and biomass, soil physicochemical properties, and enzyme activities were measured. The results showed that all three AMF treatments significantly enhanced the growth, plant biomass, and soil enzyme activity of *C. migao* seedlings. Among them, *C. etunicatum* demonstrated the most effective overall promotion. Therefore, the application of AMF, particularly *C. etunicatum*, can enhance the drought resistance of *C. migao*, which supports its large-scale cultivation and offers insights for ecological restoration in semi-arid regions.

## Introduction

Global warming and rapid industrial development are increasing the frequency of extreme weather events, which subject plants in terrestrial ecosystems to abiotic stresses—including extreme drought, high temperature, and flooding—that disrupt

**Data availability statement:** All data are in the manuscript and/or supporting information files.

**Funding:** This work was supported by the Jiangsu Education Department Technology Program (24KJD360004).

**Competing interests:** The authors have declared that no competing interests exist.

key physiological processes such as photosynthesis, osmoregulation, and membrane lipid peroxidation. [1–2]. Although water shortage impairs plant growth and productivity via multiple pathways [3–4], the limited self-protective capacity of plants is compensated by rhizosphere microorganisms, which play a key role in plant adaptation to stressful environments [5]. Due to their sensitivity to climatic and edaphic factors, these microorganisms also reflect ecosystem stability and function. Consequently, investigating the rhizosphere interface—where plant and microbial responses converge—is imperative for advancing predictive frameworks of plant performance under environmental change.

Arbuscular mycorrhizal fungi (AMF), an ancient endophytic fungal group, exist widely in nature and can form symbiotic relationships with more than 90% of plants in terrestrial ecosystems [6]. Previous studies [7–9] have demonstrated that AMF can establish a bidirectional reward relationship with plants, AMF form a symbiotic relationship with host plants in which the plant allocates 15–20% of its photosynthetic products to support fungal growth, thereby enriching the soil microbial environment. In return, the extensive hyphal network of AMF expands the root absorption area, enhancing nutrient uptake, improving soil structure, increasing photosynthetic efficiency, and ultimately promoting plant biomass accumulation. This discovery provides new ideas for the enhancement of plant stress resistance in harsh environments. Accumulating evidence indicates that AMF enhance water and mineral uptake in terrestrial plants by modifying root morphology and the rhizosphere environment [10–13]. Additionally, they alleviate drought stress through multiple pathways, including the regulation of drought-responsive genes, improved photosynthetic efficiency, and enhanced antioxidant and osmoregulatory capacity in host plants.

*Cinnamomum migao* (*C. migao*) is a tree species used for vegetation restoration and medicinal resource: its fruit extracts, containing diverse active components, are applied clinically for cardiovascular, neoplastic, and inflammatory disorders [14–16]. Unfortunately, in recent years, due to excessive and unregulated logging and impaired natural regeneration, the wild resource reserve of *C.migao* has declined rapidly. Its slow natural regeneration cannot meet the rising market demand, resulting in resource shortages that must be addressed through large-scale artificial propagation [17–18]. Therefore, how to improve the growth of *C. migao* under drought stress is a critical issue that urgently needs to be addressed.

## Materials and methods

### Experimental sites and plant materials

**Experimental sites and Field collection permits.** we carried out the field experiment using a transferred agricultural land designed for large-scale cultivation of *C.migao* as the test site. The aim was to select the drought-resistant mycorrhizal seedlings, provide a reliable supply for large-scale planting.

Our study site was located on private land that is not within any protected area, and we obtained verbal permission from the landowner prior to conducting the research. Therefore, no specific permit was required for field access.

We collected vigorous seeds of consistent size from Qiannan Prefecture, Guizhou Province, China (25°640′ N, 107°70′ E, 790.28 m a.s.l.), the native habitat of *C. migao*. an area with a subtropical monsoon climate with an average annual temperature of 16.7 °C, average annual precipitation of 259 mm, and average annual sunshine duration of 1316.9 h.

**Plant materials.** *C. migao* seeds were surface-sterilized in 5% NaClO solution for 10 min and subsequently rinsed five times with sterile water. The seeds were then sown in 200 g of sterilized sand and placed in a climate chamber (day/night, 25°C/20°C, 80% r.h.) and were irrigated with sterile water once a day. After 4 months, healthy *C. migao* seedlings with consistent growth were selected as experimental seedlings.

The experimental site has four distinct seasons, with relatively warm winters and cool summers. Eight pools with the same scale were built in the plastic greenhouse (2.8 m × 2.8 m × 0.72 m) for the experiment. The bottom was covered with waterproof boards, overlaid with bricks and concretes, and finally filled with experimental soils. The soils were collected from the in-situ field and then sieved at 2 mm. Sloped outfalls were designed for each pool to prevent the accumulation of water. The in-situ soil physicochemical properties were as follows: pH value (soil:water = 1:5), 7.03; soil organic carbon, 24.06 g·kg$^{-1}$; total nitrogen, 3.30 g·kg$^{-1}$; total phosphorus, 0.99 g·kg$^{-1}$; and total potassium, 16.02 g·kg$^{-1}$.

## AMF inoculum

Both *F. mosseae* (Accession No. YN 05) and *C. etunicatum* (Accession No. BGC GZ03C) used for AM fungal propagation were provided by the Beijing Academy of Agriculture and Forestry Science (Beijing, China). The fungi were isolated from *Sorghum bicolor* and composed of AM spores (approximately 120 spores per gram of inoculum), mycelium, sand, and root fragments. The AMF inoculant used in this experiment was propagated with *Trifolium repens* in sterilized river sand. The propagation lasted for 4 months in an artifical climate chamber. The final inoculant obtained in the propagation contained a rhizosphere soil mixture of mycelium and root fragments of infected plants, with 70 spores per gram.

## Experimental design

The experimental design comprised two factors, AMF species (including two single inoculation treatments with *F. mosseae* or *C. etunicatum*, one co-inoculation treatment with a mixture of *F. mosseae* and *C. etunicatum*, and one controlled treatment) and soil water regimes (well-watered and drought-stress).To prevent the unexpected death of the seedlings that would influence the experiment, we prepared 50 replicates (seedlings) per treatment, In total, there were 400 seedlings. The seedlings were transplanted into test pools with 20 g (dry wt) mycorrhizal with nearly 1400 spores (*F. mosseae* or *C. etunicatum* or Mixed evenly) inoculum or sterilized inoculum No fertilization was performed during the entire experimental period. The experiment consisted of three stages. In the first stage, all seedlings were well watered for 120 days (prior stress, PS) to ensure adequate colonization of AMF. Then they were subjected to drought stress (SS) for the next 30 days (seedlings began wilting), and then rewatered (REC) for the next 30 days (seedlings resumed growth). The experimental soil was supplied with water to maintain a relative water content of 75% under the well-watered treatment. Under the drought stress treatment, no water was applied until the seedlings began to wilt. There were five replicates of each treatment at each experiment stage.

## Determination of parameters

**AMF colonization rate.** After AMF inoculation for 120 days, five seedlings were randomly selected, and the roots were collected to determinate the colonization rate. The roots were flushed with sterilized water three times and then cut into 1 cm fragments. The fragments were first cleared with 10% KOH for 30 minutes at 90°C, acidified with 1% HCl for 15 minutes, and then preserved in 5% lactoglycerol solution. For analysis, the stained roots were randomly sampled and aligned parallel on optical microscope slides (CX43, Olympus, Tokyo, Japan). The AMF colonization rate was calculated according to the following formula:

$$\text{AMF colonization rate }(\%) \quad = \frac{\text{Colonized root length(cm)}}{\text{Total observed root length(cm)}} \times 100$$

 

**Root vigor.** The root vigor was measured by referring to the method of Liu, 2014 [19]. Before measurement, *C. migao* roots were randomly selected, flushed with sterilized water three times, and then cut into 1 cm fragments. The root fragments (0.5 g) were placed in a 10 ml breaker with 0.4% 2, 3, 5-triphenyltetrazolium chloride (TTC, 5 ml) and phosphate buffer solution (5 ml), kept in the dark environment for 3 hours at 37°C, and then added with 1 mol·L$^{-1}$ sulfuric acid (2 ml) to terminate the reaction. The root fragments were transferred from the 10 ml breaker to a mortar with ethyl glyoxalate solution (4 ml) and quartz sand, poured into a test tube, and washed the residue three times, and the absorbance was measured using an ultraviolet spectrophotometer at a wavelength of 485 nm. Root vigor was calculated according to the following formula:

$$\text{Reducing strength of TTC} = \frac{\text{Reduction of TTC(mg)}}{\text{Weight of root fragment(g)} \times \text{Time(h)}}$$

**Plant growth index and biomass.** Plant heights were measured with a tape measure as the distance from the base of the seedling main stem to the top. The ground diameter was measured with a Vernier caliper by determining the smoothness of the seedling 1 cm from the soil surface, and leaf areas were measured with a handheld Leaf Area Meter (YMJ-B, Top, Hangzhou, China) using five randomly selected *C. migao* seedlings. Then, the *C. migao* seedlings were separated from the soil, cleared with running water, and rinsed three times with distilled water. The seedlings were divided into root, stem, and leaf parts; baked in an oven at 105°C for 15 minutes; and dried at 70 ± 5°C to a constant weight. The dry masses of seeding roots, stems, and leaves were measured with a precision analytical balance (0.0001 g).

**Soil physicochemical properties.** Soil samples were collected from five randomly selected *C. migao* seedlings. The topsoil was removed to 2 cm in depth; 100 g of soil was sampled from around the plant roots, and 40 g was retained by a quartering method. The soil sample was dried naturally in the laboratory and then passed through a 2 mm sieve for the experiments. Four soil factors were measured (soil pH, total organic carbon content (TOC), total nitrogen content (TN), and total phosphorus content (TP). Soil pH was measured with an acidimeter (soil: water = 1:5). TOC was measured using the potassium dichromate oxidation method (GB 9834−1988, China). TN was measured using the indophenol blue colorimetry method (LY/T1269-1999, China), and TP was measured using the vanadium molybdate yellow colorimetric method (LY/T1270-1999, China).

**Soil enzyme activities.** Soil enzyme activities were determined using fresh soil samples stored at 4°C. Soil catalase enzyme activity (S-CAT) was measured using a soil catalase kit (Solarbio, BC0105, Beijing, China), and soil urease enzyme activity (S-UE) was measured with a Solarbio soil urease kit (Solarbio, BC0125, Beijing, China). Soil acid phosphatase enzyme activity (S-ACP) was measured with a soil acid phosphatase kit (Solarbio, BC0145, Beijing, China), and soil sucrase enzyme activity (S-SC) was measured with a soil sucrase kit (Solarbio, BC0245, Beijing, China).

## Data analysis

The AMF colonization rate, root vigor, seedling traits, soil physicochemical properties, and soil enzyme activities were analyzed using a two-way analysis of variance (ANOVA) and Tukey's HSD post hoc test in SPSS 22.0 (SPSS Inc., Chicago, IL, USA). The figures were produced using Origin 2019 (Origin Lab, Northampton, MA, USA). The correlation analyses among all determined indices at three stages and the principal component analysis (PCA) were implemented using https://cnsknowall.com/. All data are given as mean ± standard error. $P < 0.05$ was considered significant.

## Results

### AMF colonization and root vigor

No colonization was detected in the CK treatment under well-watered or drought conditions. The mycorrhizal colonization of *C. migao* was significantly affected by water regime and fungal species (Fig 1 and S1 Table, $P < 0.05$). The colonization rate of mixed AMF treatment was significantly reduced compared with that of single AMF inoculation. At the SS

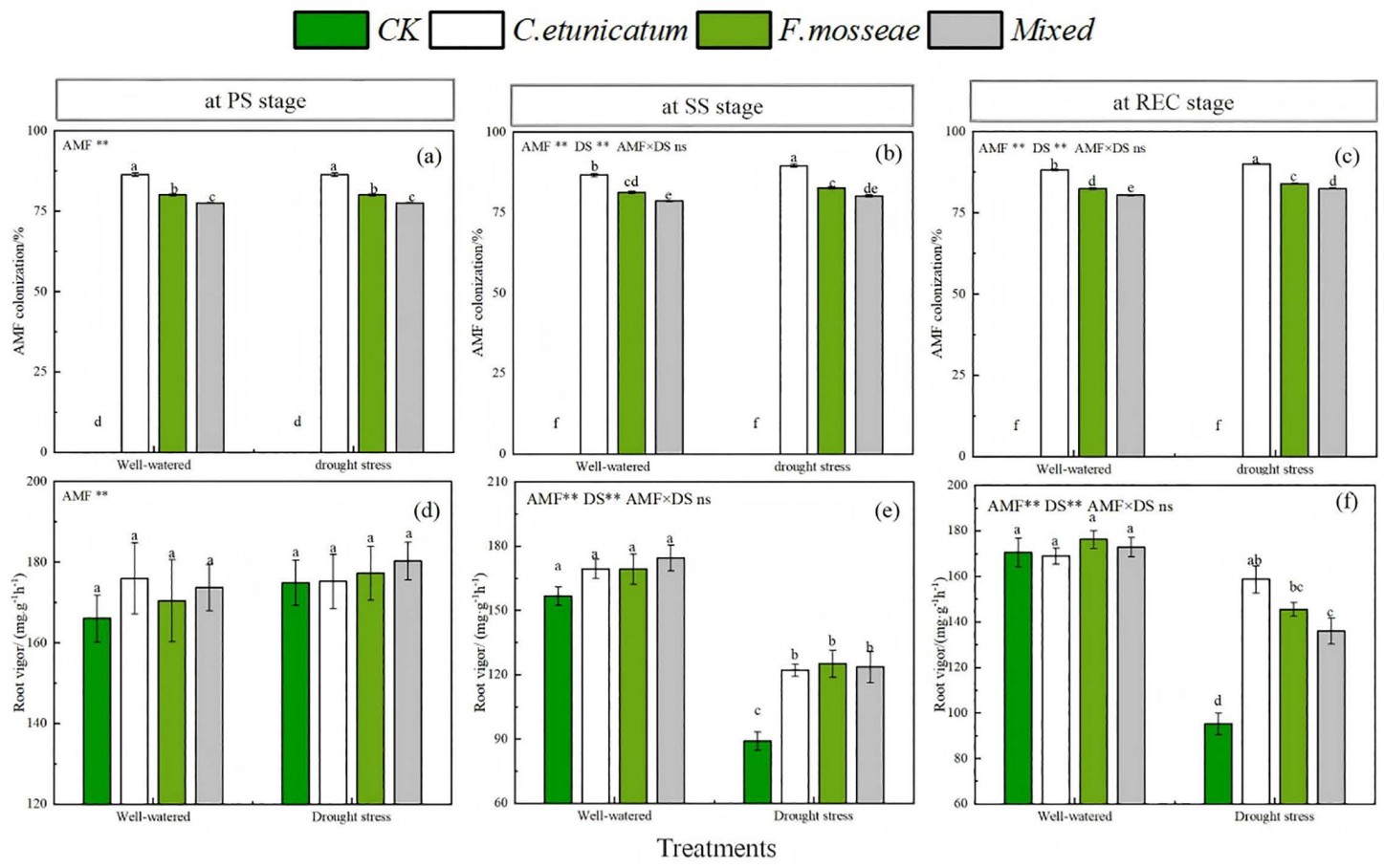

**Fig 1. AMF colonization and root vigor of *C. migao* seedlings at different growth stages.** CK, no inoculation; *C. etunicatum*, seedlings inoculated with *C. etunicatum*; *F. mosseae*, seedlings inoculated with *F. mosseae*; and *Mixed*, seedlings inoculated with *C. etunicatum* and *F. mosseae*. PS: prior stress, SS: subjected to drought stress; REC: rewatered; DS: drought stress. Different letters (a, b, c, d) indicate a significant difference by Tukey's post hoc test. ns, not significant; * $P < 0.05$ and ** $P < 0.01$. Values are expressed as the SE (n = 5, which are treatment replicates).

and REC stages, the colonization rate under drought-stress was significantly improved and was higher than that under well-watered conditions. The root vigor of *C. migao* was significantly affected by fungal species under drought-stress (Fig 1 and S1 Table, $P < 0.05$). Specifically, although there was no significant difference among all treatments in the PS stage, the root vigor of plants inoculated with AMF was significantly higher than that of those under the CK treatment in the SS and REC stages. The ANOVA results demonstrated that both the colonization rate and root vigor of *C. migao* were affected by AMF inoculation and the water regime (S1 Table).

## Plant growth indices

*C. migao* seedlings were affected by the water regime. Except for the PS stage, seedlings under drought-stress conditions showed significantly reduced plant height, stem diameter, and leaf area compared with those under well-watered conditions (Fig 2 and S1 Table, $P < 0.05$). Mycorrhizal seedlings showed significantly higher plant height, stem diameter, and leaf area than nonmycorrhizal seedlings under both well-watered and drought-stress conditions, and seedlings inoculated with *C. etunicatum* had the highest values among all treatments. The ANOVA results suggested that the plant indices mentioned above were affected by both AMF inoculation and the water regime (S1 Table).

## Plant biomasses

The results for the dry mass of mycorrhizal and nonmycorrhizal seedlings were similar to those for the plant growth indices. Dry masses of *C. migao* leaves, stems, and roots were increased by AMF inoculation and were significantly greater than those of nonmycorrhizal seedlings (Fig 3 and S1 Table, $P<0.05$). Excluding the PS stage, seedlings inoculated with *C. etunicatum* had the highest values. In the CK treatment, no significant difference was observed between well-watered and drought-stress conditions. However, seedlings under drought-stress accumulated more biomass than those under well-watered conditions. The ANOVA results suggested that inoculation with AMF augmented plant biomass among the three stages and that the water regime influenced biomass at the SS and RES stages. The interaction was significant only for leaf dry mass at the RES stage (S1 Table).

## Soil physicochemical properties

AMF inoculation had significant negative effects on the soil pH of *C. migao*. Specifically, at the PS and SS stages, the pH values of soil inoculated with *F. mosseae*, *C. etunicatum*, and their combination were lower than those of uninoculated control soil under both well-watered and drought conditions (Fig 4 and S1 Table, $P<0.05$).

AMF significantly promoted the accumulation of organic carbon in the soil. The TOC in soils inoculated with *C. etunicatum* was significantly higher than that of other treatments apart from the REC stage under well-watered conditions (Fig 4 and S1 Table, $P<0.05$). Interestingly, there was little difference between the CK treatment and the mixed inoculation treatment, regardless of the water regime.

The effects of AMF on TN in soil varied among the water regimes. Under drought-stress conditions, the TN content was decreased by the mixed inoculation treatment but was increased by the two single inoculation treatments compared with CK (Fig 4 and S1 Table, $P<0.05$). However, under well-watered conditions, the mixed inoculation treatment only reduced the TN content at the SS stage.

Drought-stress had a negative effect on phosphorus accumulation in the soil. Except for the mixed inoculated treatment, TP in soil was higher at the PS stage than at the two other stages (Fig 4 and S1 Table, $P<0.05$). For inoculated fungal species, the TP content of plants inoculated with *C. etunicatum* had the highest values regardless of the water regime.

The ANOVA results suggested that the presence of AMF had a significant effect on measured soil physicochemical properties during the experiment. There were significant effects of the water regime on the soil physicochemical properties. However, the interaction between AMF inoculation and water regime only significantly affected soil TOC at the REC stage and soil TN at the SS stage.

## Soil C:N:P ratio

The C:N:P ratio in *C. migao* soils was altered by the various treatments (Fig 5 and S1 Table). The C:N ratio of soil inoculated with AMF at the SS and REC stages was significantly influenced by the water regime and the interaction with AMF at the REC stage. Under well-watered conditions, the soil C:P ratio was increased by inoculation with AMF compared with the CK treatment. Meanwhile, the ANOVA results demonstrated that the soil C:P ratio was affected by AMF, the water regime, and their interaction during the latter two stages. The effect of AMF on the soil N:P ratio was observed throughout the experiment. The water regime only influenced the soil N:P ratio at the SS stage.

## Soil enzyme activities

Both AMF inoculation and water regime had a positive effect on soil enzyme activities compared with the CK treatment (Fig 6 and S1 Table, $P<0.05$). Among all treatments, S-UE and S-SC activities had the highest values in the treatment with *C. etunicatum*, and there were similar results for S-CAT and S-ACP under well-watered conditions. In contrast, under drought-stress conditions, S-CAT obtained its highest values in the treatment with *F. mosseae* at the SS stage and in the

 

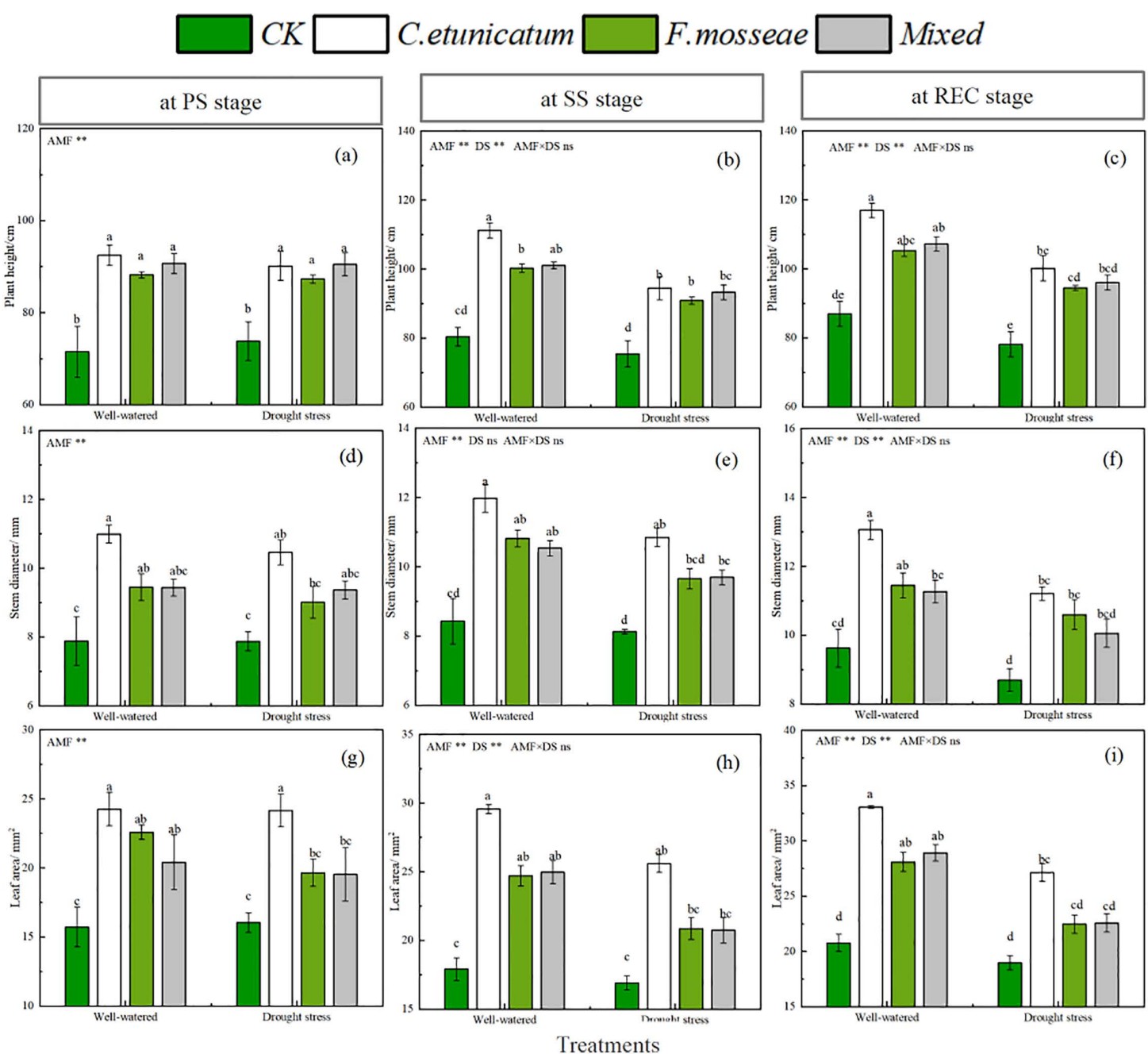

**Fig 2. Growth of *C. migao* seedlings at different growth stages.** CK, no inoculation; *C. etunicatum*, seedlings inoculated with *C. etunicatum*; *F. mosseae*, seedlings inoculated with *F. mosseae*; and *Mixed*, seedlings inoculated with *C. etunicatum* and *F. mosseae*. PS: prior stress, SS: subjected to drought stress; REC: rewatered; DS: drought stress. Different letters (a, b, c, d) indicate a significant difference by Tukey's post hoc test. ns, not significant; * $P < 0.05$ and ** $P < 0.01$. Values are expressed as the SE (n = 5, which are treatment replicates).

mixed inoculation treatment at the REC stage. At the SS stage, the activity of S-CAP was slightly higher in the mixed inoculation treatment than in other treatments. The ANOVA results demonstrated that both AMF and water regime significantly affected soil enzyme activities.

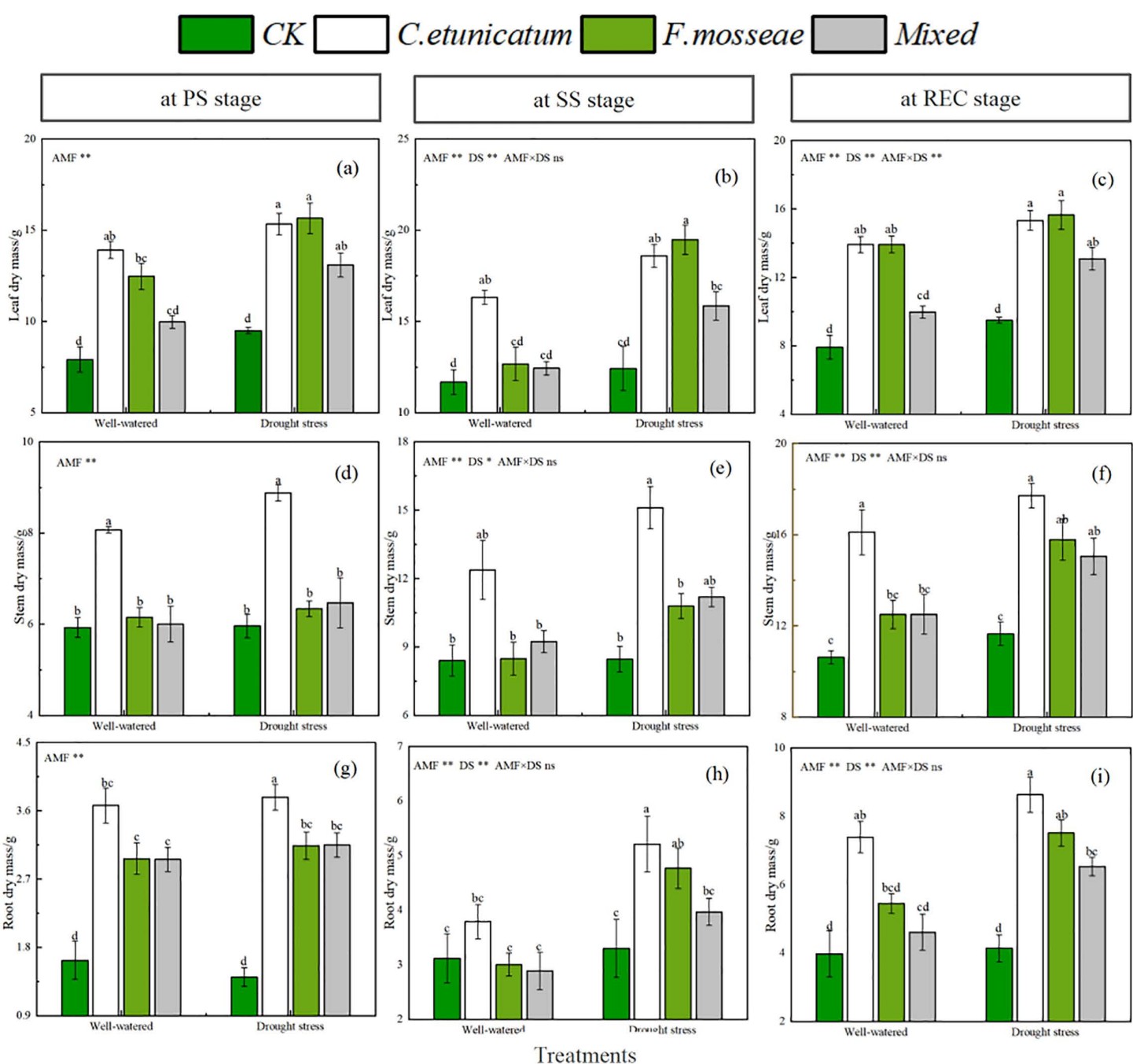

**Fig 3. Plant biomasses of *C. migao* seedlings at different growth stages.** CK, no inoculation; *C. etunicatum*, seedlings inoculated with *C. etunicatum*; *F. mosseae*, seedlings inoculated with *F. mosseae*; and *Mixed*, seedlings inoculated with *C. etunicatum* and *F. mosseae*. PS: prior stress, SS: subjected to drought stress; REC: rewatered; DS: drought stress. Different letters (a, b, c, d) indicate a significant difference by Tukey's post hoc test. ns, not significant; * $P < 0.05$ and ** $P < 0.01$. Values are expressed as the SE ($n = 5$, which are treatment replicates).

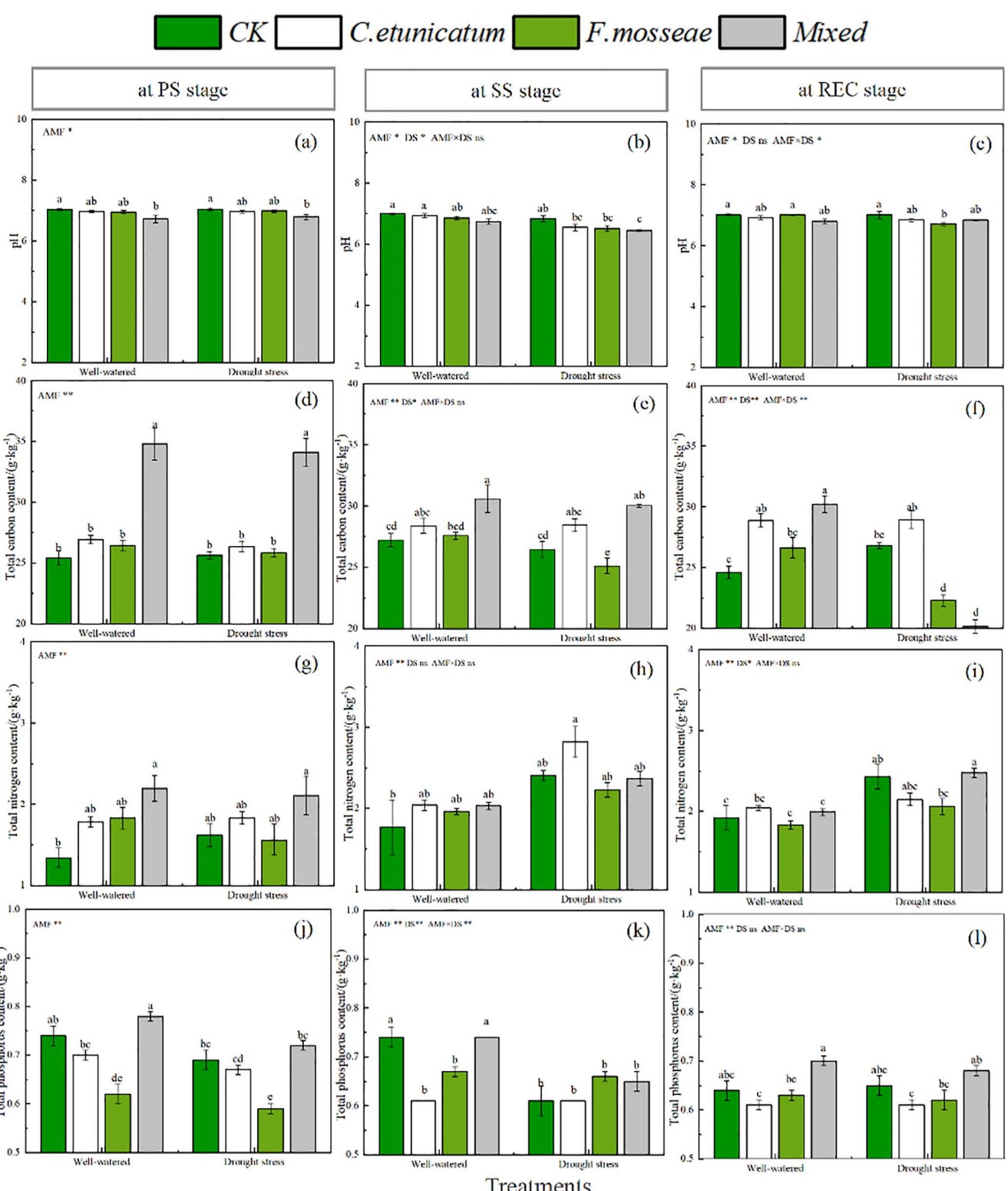

**Fig 4. Soil physicochemical properties of *C. migao* seedlings at different growth stages.** CK, no inoculation; *C. etunicatum*, seedlings inoculated with *C. etunicatum*; *F. mosseae*, seedlings inoculated with *F. mosseae*; and *Mixed*, seedlings inoculated with *C. etunicatum* and *F. mosseae*. PS: prior

stress, SS: subjected to drought stress; REC: rewatered; DS: drought stress.. Different letters (a, b, c, d) indicate a significant difference by Tukey's post hoc test. ns, not significant; * $P < 0.05$ and ** $P < 0.01$. Values are expressed as the SE (n = 5, which are treatment replicates).

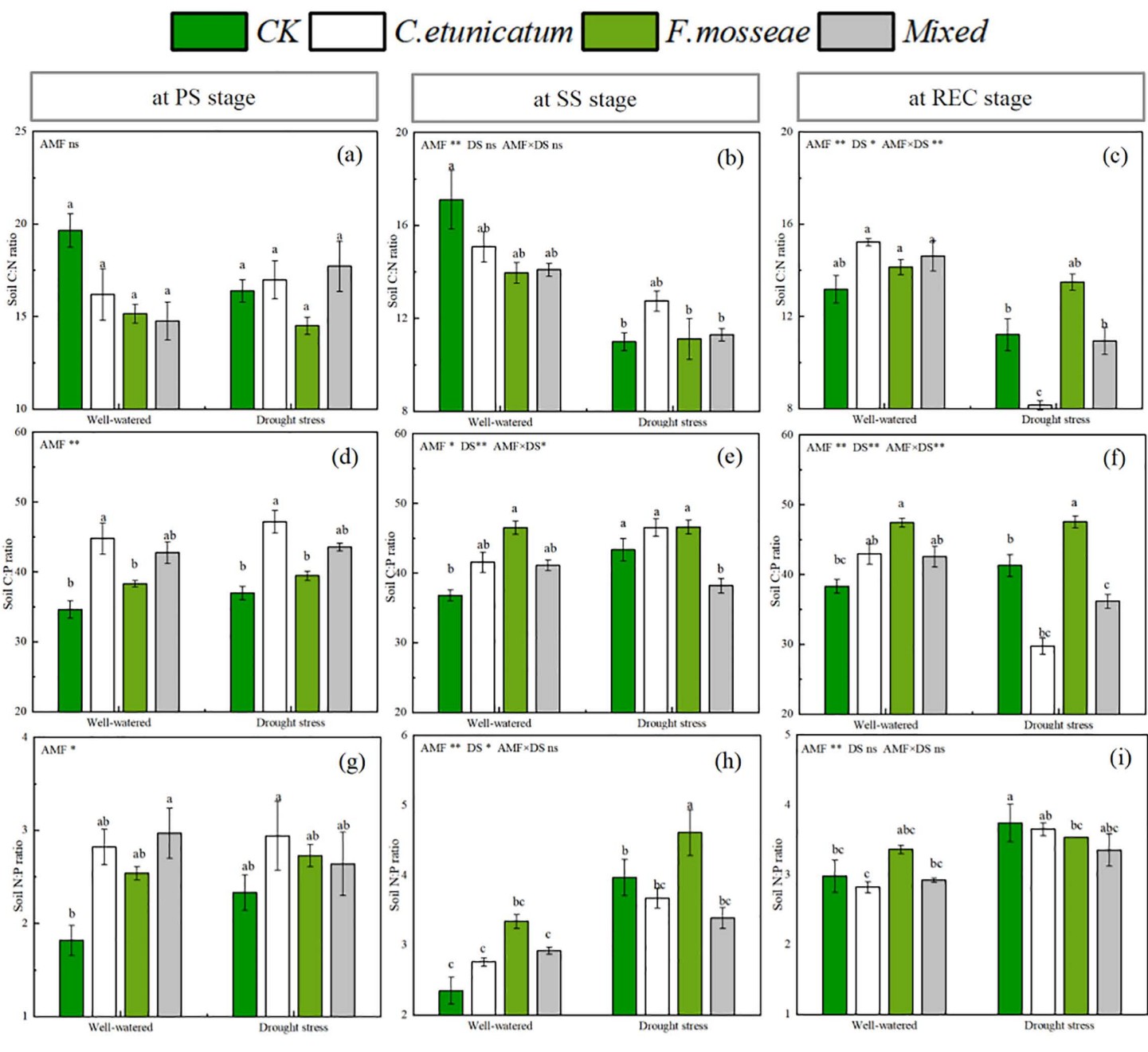

**Fig 5. Soil C:N:P ratios of *C. migao* seedlings at different growth stages.** CK, no inoculation; *C. etunicatum*, seedlings inoculated with *C. etunicatum*; *F. mosseae*, seedlings inoculated with *F. mosseae*; and *Mixed*, seedlings inoculated with *C. etunicatum* and *F. mosseae*. PS: prior stress, SS: subjected to drought stress; REC: rewatered; DS: drought stress. Different letters (a, b, c, d) indicate a significant difference by Tukey's post hoc test. ns, not significant; * $P < 0.05$ and ** $P < 0.01$ Values are expressed as the SE (n = 5, which are treatment replicates).

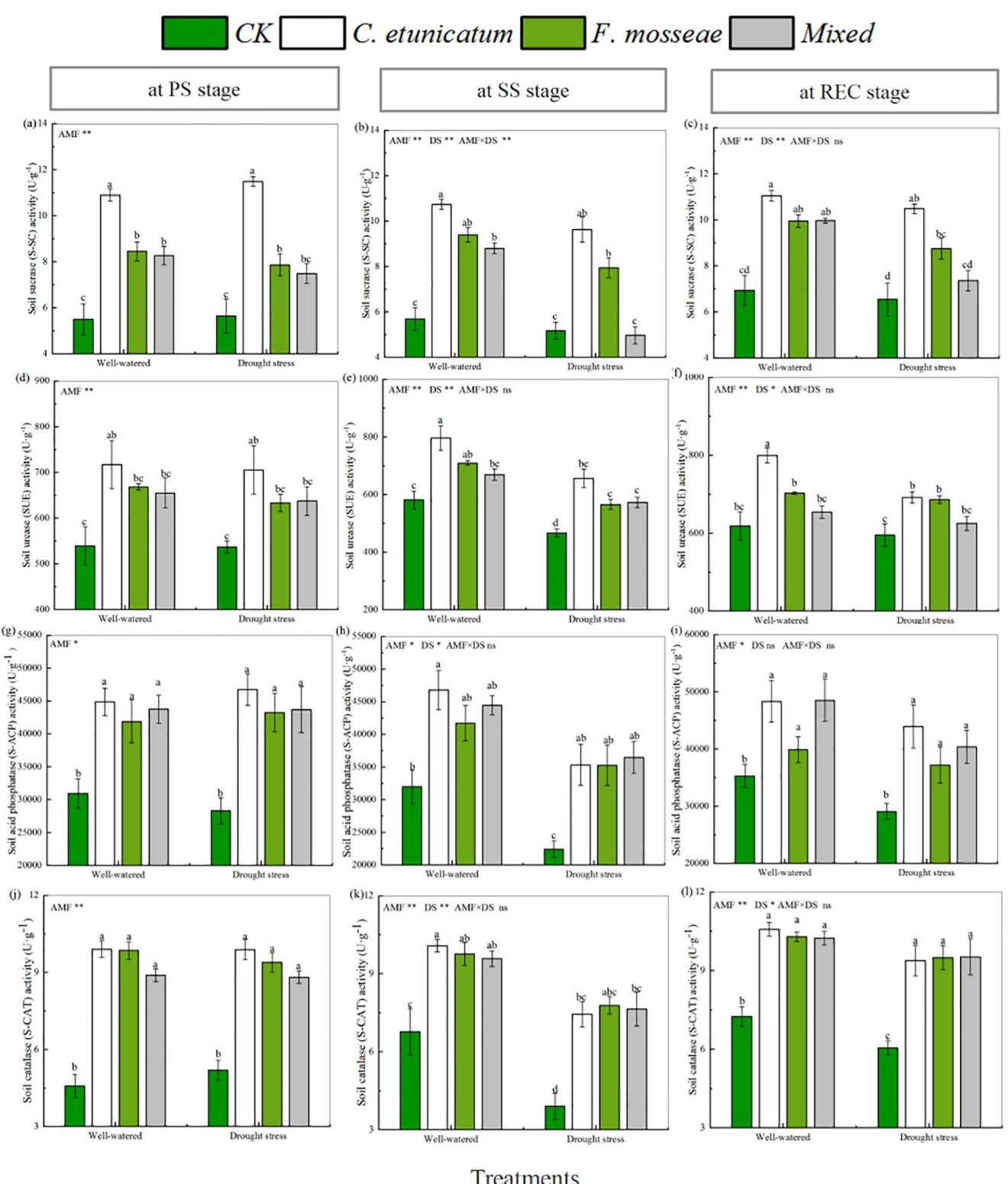

**Fig 6. Soil enzyme activities of *C. migao* seedlings at different growth stages.** CK, no inoculation; *C. etunicatum*, seedlings inoculated with *C. etunicatum*; *F. mosseae*, seedlings inoculated with *F. mosseae*; and *Mixed*, seedlings inoculated with *C. etunicatum* and *F. mosseae*. PS: prior stress,

## Correlation analysis

We examined the correlations between plant growth indices and biomass, soil physicochemical properties, and enzyme activities at different stages. At the PS stage (Fig 7a and S1 Table), all of the parameters showed positive correlations except for soil TP, pH, and the C:N ratio. Soil TP was negatively correlated with plant height (r = −0.107, P = 0.512), C:P ratio (r = −0.195, P = 0.227), and N:P ratio (r = −0.223, P = 0.167). Both soil pH and the C:N ratio were negatively related to the three measured plant growth indices, four soil enzyme activities, and two soil characters (TN and TOC). In addition,

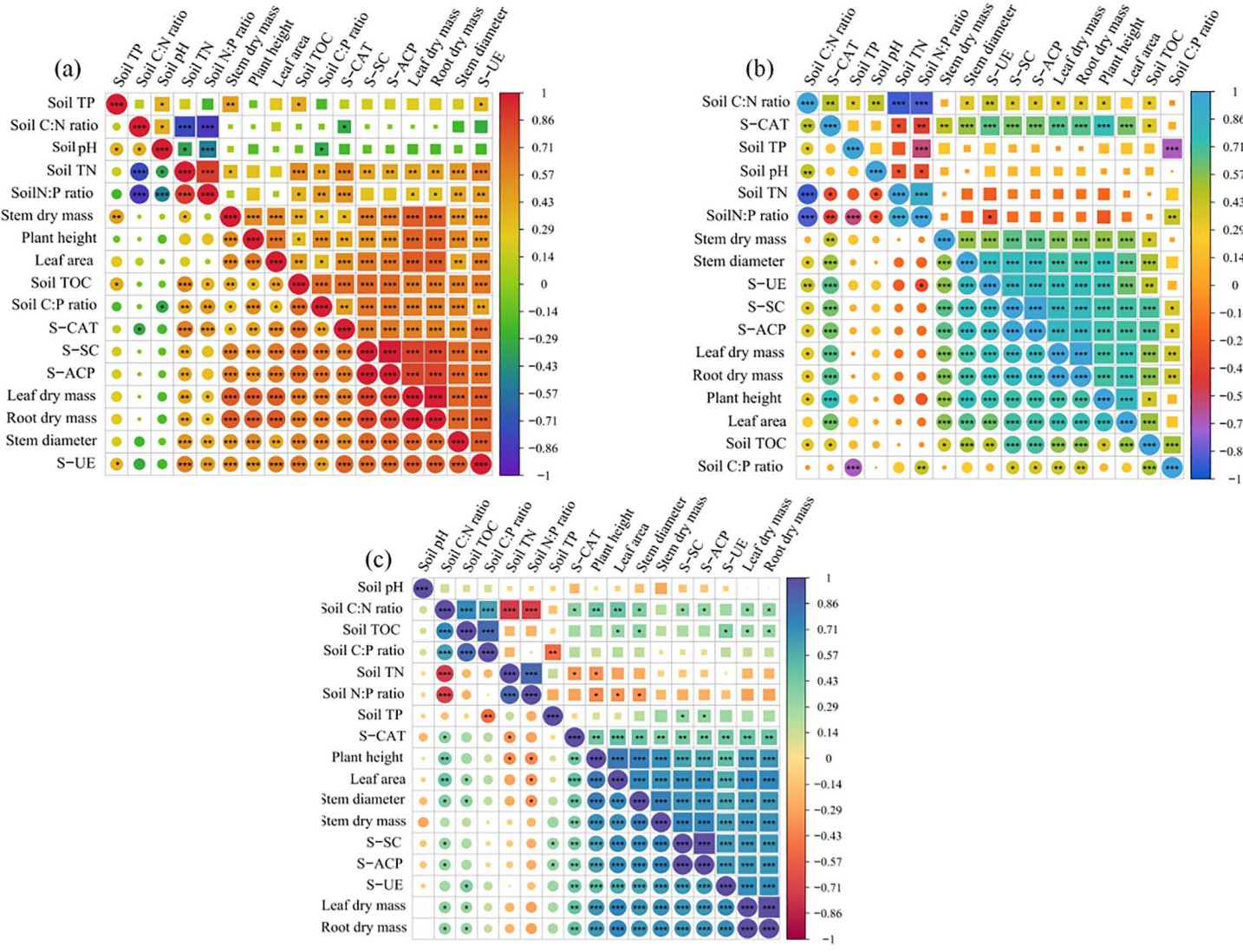

**Fig 7. Correlation analysis at different growth stages.** * $p < 0.05$: significant correlation.

there were strong positive correlations among the plant growth indices, plant biomass, and enzyme activities at the PS stage.

Drought stress affected the original correlations among the measured indicators. At the SS stage (Fig 7b and S1 Table), soil TP was positively related to plant height and negatively related to soil TN, leaf dry mass, and root dry mass, in contrast to the results from the PS stage. Significant changes were also observed in soil TN, pH, and the C:N:P ratio. Soil TN was positively correlated with stem dry mass, soil TOC, and the C:P and N:P ratios and negatively correlated with other measured indicators. Concurrently, soil pH was positively correlated with plant growth indices, plant dry mass, soil enzyme activities, and the two soil characters of TN and TP.

At the REC stage (Fig 7c and S1 Table), there were positive correlations among plant growth indices, dry mass, and soil enzyme activities, consistent with the results at the above two stages. Both soil TN and the N:P ratio were negatively related to plant growth indices, dry mass, and soil enzyme activities. Soil pH showed positive correlations with leaf and root dry mass, soil TOC, the C:N ratio, and the C:P ratio, but negative correlations with other parameters.

### Principal component analysis (PCA)

At the PS stage (Fig 8a and S1 Table), principal component 1 (PC1) and principal component 2 (PC2) accounted for 53.5% and 16.7% of the variance, respectively. Compared with water regimes, samples from the different AMF inoculation treatments were further apart on the PC1 axis, suggesting a greater effect of the AMF species.

Changes occurred at the SS (Fig 8b and S1 Table) and REC (Fig 8c and S1 Table) stages, where the PCA plot revealed that the first two components explained 67.1% (PC1, 46.9%; PC2, 20.2%) and 57.5% (PC1, 39.7%; PC2, 17.8%) of the variance. The samples were relatively dispersed both under various water conditions and different AMF inoculation treatments. This indicated that significant variation existed in the samples under different treatments, including water regimes, AMF species, and their interaction.

## Discussion

As one of the major abiotic limitations to plant growth, water deficit is responsible for soil degradation and thus has significant effects on global crop yields [20–21]. Climate extremes and man-made disruptions have led to changes in precipitation, increasing the severity and duration of droughts [22–23]. Currently, the majority of studies have focused on the tolerance and adaptation mechanisms of plants to drought stress [24–26]. In the present study, we investigated the effects of the AMF *C. etunicatum* and *F. mosseae* and their combination in *C. migao* seedlings under both drought-stress and well-watered conditions.

Whether water deficit is conducive to promoting the formation of symbionts in the plant rhizosphere such as AMF has been controversial. Some previous studies suggested a negative role of drought in the colonization of the plant rhizosphere by AMF [27–29]. However, in the present study, drought stress significantly promoted colonization. *C. migao* seedlings under drought-stress conditions had higher colonization rates than under the well-watered conditions, similar to the findings of Xie et al. [30]. The reasons for the opposite results involved differences in the tolerance and the ability of various fungal and plant species to respond to environmental pressure [19]. Among the treatments, seedlings inoculated with *C. etunicatum* had the highest colonization rate due to the strong adaptability of this species to harsh environments (such as water and nutrient deficits), consistent with the results of the pot experiment by Xiao et al [31].

Fine roots serve as the key organ connecting the external environment and terrestrial plants. Plant life activities can be objectively assessed by the root vigor [32–33]. However, the increased possibility of plant roots being damaged by water deficit hinders nutrient absorption and substance transformation by plants, suggesting that drought stress will negatively affect plant root growth [34]. This was consistent with our finding that drought reduced root vigor in *C. migao* seedlings. Previous research has demonstrated that invasive mycorrhizal fungi could induce related growth hormone production and promote soil nutrient absorption, thereby enhancing plant root vigor [35–36]. Similar to our findings, seedlings that

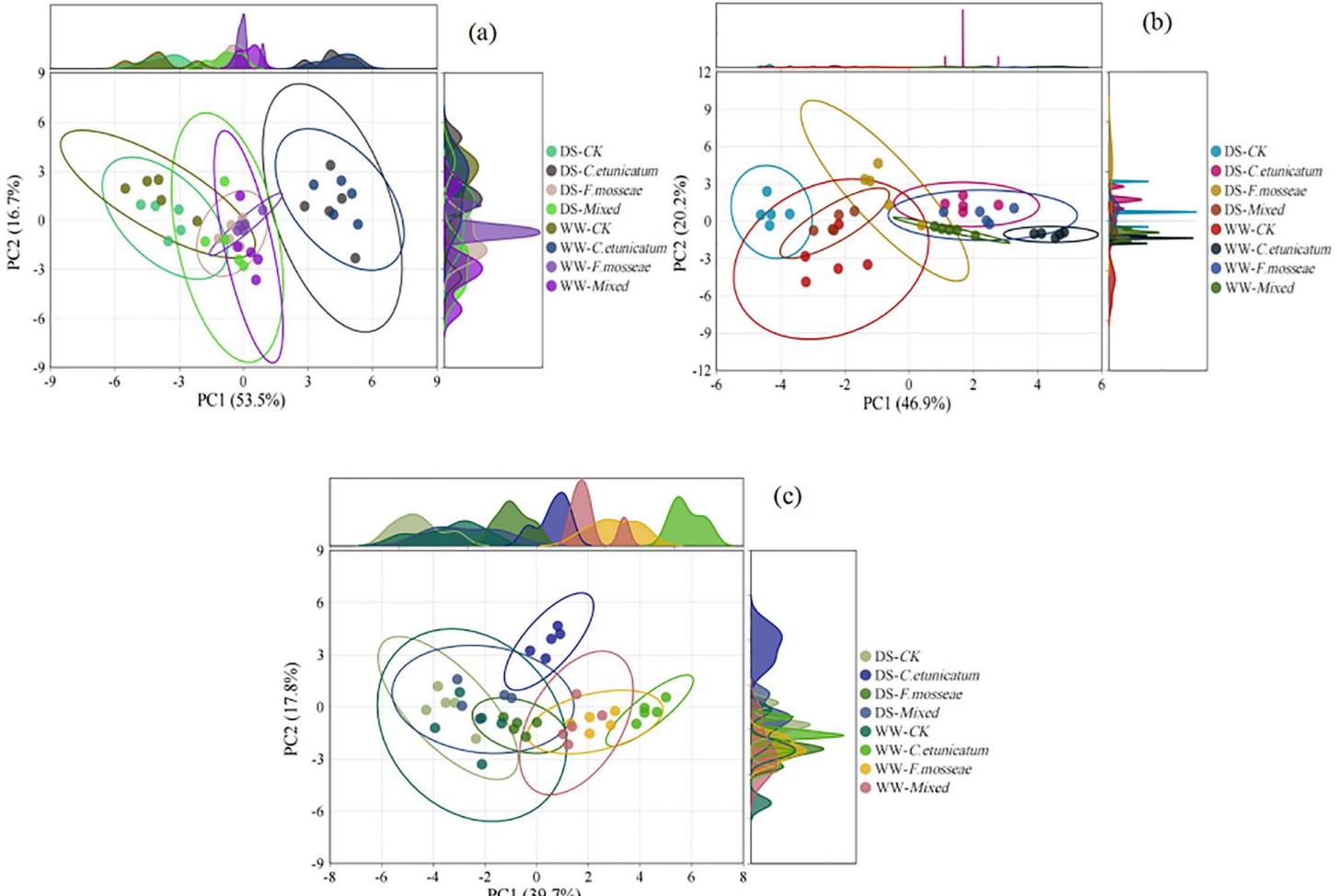

**Fig 8. PCA at different growth stages.** CK, no inoculation; *C. etunicatum*, seedlings inoculated with *C. etunicatum*; *F. mosseae*, seedlings inoculated with *F. mosseae*; and Mixed, seedlings inoculated with *C. etunicatum* and *F. mosseae*. DS: drought stress; WW: well-watered.

received AMF inoculation showed better performance than nonmycorrhizal seedlings, indicating that AMF were effective in alleviating the damage to host roots by drought stress [37].

Negative effects on plant performance under drought-stress, including plant growth, biomass accumulation, and soil nutrient cycling, have been well documented [38]. As a consequence, physiological and biochemical processes are regulated to adapt the host plants to drought. One of the strategies involves forming symbioses between the plant rhizosphere and AMF [39–40]. This has been demonstrated in research on *Hibiscus cannabinus*, *Avena sativa*, *Ammodendron bifolium*, and other plants [41–43]. In this study, *C. migao* seedlings reacted strongly to water regimes in phenotypic traits (including plant height, stem diameter, and leaf area) and dry masses at the SS and RES stages, in line with previous studies [44]. Furthermore, our findings showed that the presence of AMF promoted biomass accumulation and plant growth and that seedlings inoculated with *C. etunicatum* performed best among all treatments, demonstrating its growth-promoting function toward *C. migao* seedlings [44].

Soil microorganisms, especially AMF, participate in a series of biogeochemical processes, including organic decomposition and nutrient cycling, thereby playing an irreplaceable role in maintaining soil functions [45–47]. These effects serve

as major driving forces for sustaining plant productivity in terrestrial ecosystems, particularly in soils lacking essential mineral elements such as phosphorus or nitrogen. Additionally, networks formed by AMF at host roots are conducive to resisting drought stress, heavy metals, and pests, further maintaining the health of the local soil environment and promoting plant growth [48]. Similar to our findings, in Lahouki's study of *Cactus* [49], AMF significantly accelerated the accumulation of TOC, TN, and AP under drought conditions. Further confirmation follows from studies by Ikan et al, Madoul et al, and Xia et al [50–52].

The soil C:P:N ratio is an important indicator of soil organic mineralization and nutrient availability [53–55]. Microorganisms, especially AMF, play a crucial role in stabilizing plant community productivity through soil mineralization and nutrient transformation. Specifically, the C:N ratio reflects the mineralization rate of soil organic matter; the C:P ratio represents the potential for soil microorganisms to mineralize organic matter and release phosphorus; and the N:P ratio has key limiting effects of N and P on plant growth [56–59].

In the present study, the soil C:N ratio was lower under drought-stress conditions than under well-watered conditions; the microorganisms had a strong ability to supply N to the rhizosphere soil at this time [60]. Furthermore, the C:N ratio of <25:1 as a reasonable organic matter decomposition rate was in line with previous results [61]. Previous studies have suggested that a lower C:P ratio is beneficial for microbial nutrient uptake during organic matter decomposition and facilitates available phosphorus accumulation in the soil [62]. In our study, *C. etunicatum* inoculation alone and in combination with *F. mosseae* decreased the rhizosphere soil C:P ratio, and both inoculation treatments promoted the availability of phosphorus in the rhizosphere of *C. migao* regardless of the water regimes. Generally, when the N:P ratio is less than 14:1, plant growth is considered to be limited by N [63]. In this study, the soil N:P ratio ranged from 1.82:1 to 4.61:1, suggesting that N was one of the main factors limiting the growth of *C. migao* seedlings.

Soil pH and microorganisms engage in bidirectional regulation. While microbes influence pH through biochemical processes to facilitate nutrient absorption, pH itself structures the rhizosphere microbiome, ultimately fostering plant development [64–66].. The correlation analysis showed that soil pH was significantly associated with TP, TN, and the C:N and N:P ratios, demonstrating that nutrient balance in plant rhizosphere soil was maintained through the interaction between pH and microorganisms.

Soil extracellular enzymes from microbial metabolism are involved in organic matter decomposition and humus formation [67–69]. Enzyme activities can serve as indicators of the most sensitive responses of microbial communities to environmental stress, and the enzymes play crucial roles in biogeochemical cycling [70]. However, the positive effects mentioned above are easily interfered with by harsh environmental conditions such as water deficits, climate change, and heavy metal toxicity [68]. Fortunately, some specific AMF and plant growth-promoting rhizobacteria exist in the plant rhizosphere soil to counteract such stresses [71]. Previous research has demonstrated that AMF inoculation could significantly enhance enzyme activities, including those of S-UE, S-ACP, and S-SC, at the host plant rhizosphere, as shown by the significantly positive correlation with accumulated plant biomass [71–73]. Similar results were observed in our research; although four measured enzyme activities were weakened under drought-stress conditions, Enzyme activities were significantly higher in seedlings inoculated with AMF than in the non-inoculated control group. Meanwhile, as demonstrated by the correlation analysis, the activities of S-SC, S-ACP, S-UE, and S-CAT were significantly positively correlated with root, stem, and leaf dry biomass of *C. migao* seedlings at three stages.

## Conclusion

We evaluated the effects of *F. mosseae*, *C. etunicatum*, and their combined inoculation on the growth, soil physicochemical properties, and enzyme activities of *C. migao* seedlings under both well-watered and drought-stress conditions. The results showed that selected fungi colonizing the plant rhizosphere can form symbioses with *C. migao* seedlings and AMF facilitated plant growth and soil nutrient acquisition. *C. etunicatum* was the more effective species for improving plant growth and nutrient uptake among all treatments, regardless of the water regime. In the future, we should conduct more

research on the biological characteristics of each type of AMF to understand its ecological response and impact under drought stress. And further research on the changes in plant hormones and signal transduction substances mediated by AMF.

## Supporting information

**S1 Table. Statistical results and original data for the figures in the manuscript.**
(DOC)

## Acknowledgments

We would like to thank the LetPub (www.letpub.com.cn) for its linguistic assistance during the preparation of this manuscript.

## Author contributions

**Conceptualization:** Xiao Xuefeng, Hao Gang, Xu Lu.

**Formal analysis:** Tian Xiu, Xu Lu, Huang Rui.

**Funding acquisition:** Hao Gang, Huang Rui.

**Investigation:** Xiao Xuefeng.

**Methodology:** Tian Xiu, Hao Gang.

**Supervision:** Hao Gang.

**Validation:** Xiao Xuefeng, Hao Gang.

**Visualization:** Huang Rui, Zan Yue.

**Writing – original draft:** Xiao Xuefeng.

**Writing – review & editing:** Tian Xiu.

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
