## [Decision Letter · Decision Letter 0]

8 Nov 2025

Arbuscular mycorrhizal fungi improve drought toleration in Cinnamomum migao H.W.Li seedlings by increasing plant growth, nutrient uptake and biomass accumulation

PLOS ONE

Dear Dr. Hao,

Thank you for submitting your manuscript to PLOS ONE. After careful consideration, we feel that it has merit but does not fully meet PLOS ONE’s publication criteria as it currently stands. Therefore, we invite you to submit a revised version of the manuscript that addresses the points raised during the review process.

We look forward to receiving your revised manuscript.

Kind regards,

Debasis Mitra

Academic Editor

PLOS ONE

Journal Requirements:

“We would like to thank the Jiangsu Education Department Technology Program,“Effects of different Arbuscular Mycorrhizal Fungi on growth and active ingredient accumulation of Bletilla striata under drought stress” (24KJD360004) for the financial support.”

Please state what role the funders took in the study.  If the funders had no role, please state: 'The funders had no role in study design, data collection and analysis, decision to publish, or preparation of the manuscript.'

Reviewer's Responses to Questions

**Comments to the Author**

1. Is the manuscript technically sound, and do the data support the conclusions?

Reviewer #1: Yes

Reviewer #2: Yes

2. Has the statistical analysis been performed appropriately and rigorously?

Reviewer #1: Yes

Reviewer #2: Yes

3. Have the authors made all data underlying the findings in their manuscript fully available?

Reviewer #1: Yes

Reviewer #2: No

4. Is the manuscript presented in an intelligible fashion and written in standard English?

Reviewer #1: Yes

Reviewer #2: Yes

Reviewer #1: The manuscript “Arbuscular mycorrhizal fungi improve drought toleration in Cinnamomum migao H.W.Li seedlings by increasing plant growth, nutrient uptake and biomass accumulation” is scientifically sound, and presents novel and relevant findings on the role of arbuscular mycorrhizal fungi in improving drought tolerance of Cinnamomum migao seedlings. The experimental design is appropriate, the data are comprehensive, and the conclusions are well supported.

Some minor revisions are required, mainly to improve clarity in language, refine certain methodological details, and strengthen the discussion with more concise mechanistic insights.

1. Abstract: Revise the first sentence “To select drought-resistant mycorrhizal seedlings, provide a reliable supply of young plants for large-scale planting” for clarity. Also add quantitative highlights (% increase in biomass, enzyme activity).

2. Introduction: It is well written, but some sentences are repetitive (climate change effects are mentioned multiple times). Some specie-specific studies on on Lauraceae or rare medicinal trees can be cited to strengthen the justifications.

3. Materials and methods: Include the spore density used for inoculation (you mention 20 g inoculum but not final spore numbers per plant). In the drought stress design, was the water potential or relative water content of soil measured? Additionally, indicate the replicates (n per treatment for final analyses).

4. Result and discussion: Some references are generic; consider integrating more recent AMF–drought studies in medicinal or woody plants.

Reviewer #2: The manuscript explores the effects of arbuscular mycorrhizal (AM) fungi on drought tolerance in Cinnamomum migao seedlings. The topic is relevant and aligns with PLOS ONE’s scope. The experimental design and results appear valid and contribute valuable insights into plant–microbe interactions and ecological restoration. Overall, the study is scientifically sound.However, the manuscript requires minor revision to improve clarity, structure, and readability. The language and grammar need careful editing. Most of the issues identified are preliminary and organizational and do not affect the scientific validity of the work.The Abstract includes repetitive, long, and complex sentences. Please define abbreviations (e.g., “AM fungi”) at first mention and focus on key findings rather than extended background information.The Introduction provides adequate background but includes methodological details that should be moved to the Materials and Methods section. It should be summarized and clearly state the research objectives.The Materials and Methods section is generally clear but requires minor clarification. Define all abbreviations, specify sample sizes, and indicate measurement units for all parameters.The Results are informative but need better organization under clear subheadings and with clearer figures. Define symbols, add units for all parameters, and ensure figure consistency.The Discussion is includes many long and repetitive sentences. Use different wording and condense overlapping ideas. Including an outlook on future research would strengthen this section.In the References section, it is important to review and apply PLOS ONE’s Vancouver style.

.

Reviewer #1: No

Reviewer #2: No

---

## [Author Response · Author response to Decision Letter 1]

1 Mar 2026

Dear professors:

Re: Manuscript ID: PONE-D-25-45963 and Title: Arbuscular mycorrhizal fungi improve drought toleration in Cinnamomum migao H.W.Li seedlings by increasing plant growth, nutrient uptake and biomass accumulation.

Hope this email finds you well.

We deeply appreciate your positive evaluation of our work. We have studied your comments carefully, which are very helpful for revising and improving our paper, as well as the important guiding significance to other research. We have carefully considered all comments and revised the manuscript accordingly. Below, we provide point-by-point responses to your comments (revisions in the manuscript are marked in yellow, and responses are highlighted in blue). Thank you very much for your attention and consideration!

Reviewer #1:

Question 1: Abstract: Revise the first sentence “To select drought-resistant mycorrhizal seedlings, provide a reliable supply of young plants for large-scale planting” for clarity. Also add quantitative highlights (% increase in biomass, enzyme activity).

Response: As suggested by the reviewer, we have revised the abstract accordingly. The updated version is provided below.

“Drought stress is a primary factor reducing field crop productivity, and its impact is predicted to intensify and occur more often because of human-influenced environmental and climate changes. Which exerts a critical influence on plant growth and distribution, especially in semi-arid Karst regions including southwest China. Cinnamomum migao H.W.Li (C. migao), a tree in the Cinnamomum genus of Lauraceae family, is a medicinally important tree species endemic to southwest China. Arbuscular mycorrhizal fungi (AMF) symbiosis mitigates drought stress in plants, yet the inoculation method affects the establishment and function of this symbiosis remains unclear. Therefore, we conducted an experiment examining the influence of different AMF (Funneliformis mosseae (F. mosseae) and Claroideoglomus etunicatum (C. etunicatum) their combination (Mixed)) on C. migao seedlings. AMF colonization rates, root vigor, seedling growth and biomass, soil physicochemical properties, and enzyme activities were measured. The results showed that all three AMF treatments significantly enhanced the growth, plant biomass, and soil enzyme activity of C. migao seedlings. Among them, G. etunicatum demonstrated the most effective overall promotion. Obviously, the application of AMF, particularly G. etunicatum, can enhance the drought resistance of C. migao, which supports its large-scale cultivation and offers insights for ecological restoration in semi-arid regions.”

Question 2: Introduction: It is well written, but some sentences are repetitive (climate change effects are mentioned multiple times). Some specie-specific studies on Lauraceae or rare medicinal trees can be cited to strengthen the justifications.

Response: Thank you for your professional advice, we have cited some newest studies or rare medicinal trees with your recommendations.

“Gu L, Schumacher DL, Fischer EM, Slater LJ, Yin J, Sippel S, Chen J, Liu P, Knutti Ret al. Flash drought impacts on global ecosystems amplified by extreme heat. Nat Geosci. 2025, 2:1–7. https://doi:10.1038/s41561-025-01719-y

Van den Bosch M, Costanza JK, Peek RA, Mola JM, Steel ZL. Climate change scenarios forecast increased drought exposure for terrestrial vertebrates in the contiguous United States. Commun Earth Environ. 2024, 5(1):708. https://doi.org/10.1038/s43247-024-01880-z

Khan, A.G., Naz, H. Role of Arbuscular Mycorrhizal Fungi (AMF) in the Production of Medicinal Crops. In: Parihar, M., Rakshit, A., Adholeya, A., Chen, Y. (eds) Arbuscular Mycorrhizal Fungi in Sustainable Agriculture: Nutrient and Crop Management. 2024, Springer, Singapore. https://doi.org/10.1007/978-981-97-0300-5_16

Ling H, Xu F, Shabbir I, Sulaiman Z, Shahbaz M, et al. Efficacy of peat-based bioformulation of microbial co-inoculants with silicon for growth promotion of rubber plants. PLOS ONE. 2025, 20(10): e0331899. https://doi.org/10.1371/journal.pone.0331899”

Question 3: Materials and methods: Include the spore density used for inoculation (you mention 20 g inoculum but not final spore numbers per plant). In the drought stress design, was the water potential or relative water content of soil measured? Additionally, indicate the replicates (n per treatment for final analyses).

Response: We are very sorry for our negligence of final spore numbers per plant and the replicates. Now we have re-written the Experimental design. The updated version is provided below.

“The experimental design comprised two factors, AMF species (including two single inoculation treatments with F. mosseae or C. etunicatum, one co-inoculation treatment with a mixture of F. mosseae and C. etunicatum, and one controlled treatment) and soil water regimes (well-watered and drought-stress).To prevent the unexpected death of the seedlings that would influence the experiment, we prepared 50 replicates (seedlings) per treatment, In total, there were 400 seedlings. The seedlings were transplanted into test pools with 20 g (dry wt) mycorrhizal with nearly 1400 spores (F. mosseae or C. etunicatum or Mixed evenly) inoculum or sterilized inoculum. No fertilization was performed during the entire experimental period. The experiment consisted of three stages. In the first stage, all seedlings were well watered for 120 days (prior stress, PS) to ensure adequate colonization of AMF. Then they were subjected to drought stress (SS) for the next 30 days (seedlings began wilting), and then rewatered (REC) for the next 30 days (seedlings resumed growth). The experimental soil was supplied with water to maintain a relative water content of 75% under the well-watered treatment. Under the drought stress treatment, no water was applied until the seedlings began to wilt. There were five replicates of each treatment at each experiment stage.”

Question 3: Result and discussion: Some references are generic; consider integrating more recent AMF–drought studies in medicinal or woody plants.

Response: Thank you for your professional advice, we have cited some studies in medicinal plants with your recommendations.

“Xiao X, Liao X, Yan Q, Xie Y, Chen J, Liang G, Chen M, Xiao S, Chen Y, Liu J et al. Arbuscular Mycorrhizal Fungi Improve the Growth, Water Status, and Nutrient Uptake of Cinnamomum migao and the Soil Nutrient Stoichiometry under Drought Stress and Recovery. J. Fungi. 2023, 9, 321. https://doi.org/10.3390/jof9030321

Zhao Y, Cartabia A, Lalaymia I, Declerck S. Arbuscular mycorrhizal fungi and production of secondary metabolites in medicinal plants. Mycorrhiza. 2022, 32, 221–256. https://doi.org/10.1007/s00572-022-01079-0

Shi S, Tian L, Ma, L, Tian L, et al. Community structure of rhizomicrobiomes in four medicinal herbs and its implication on growth management. Microbiology. 2018, 87, 425–436. https://doi.org/10.1134/S0026261718030098

Bothe H, Turnau K, Regvar M. The potential role of arbuscular mycorrhizal fungi in protecting endangered plants and habitats. Mycorrhiza. 2010, 20, 445–457. https://doi.org/10.1007/s00572-010-0332-4”

Reviewer #2:

Question 1: The Abstract includes repetitive, long, and complex sentences. Please define abbreviations (e.g., “AM fungi”) at first mention and focus on key findings rather than extended background information.

Response: As suggested by the reviewer, we have revised the abstract accordingly. The updated version is provided below.

“Drought stress is a primary factor reducing field crop productivity, and its impact is predicted to intensify and occur more often because of human-influenced environmental and climate changes. Which exerts a critical influence on plant growth and distribution, especially in semi-arid Karst regions including southwest China. Cinnamomum migao H.W.Li (C. migao), a tree in the Cinnamomum genus of Lauraceae family, is a medicinally important tree species endemic to southwest China. Arbuscular mycorrhizal fungi (AMF) symbiosis mitigates drought stress in plants, yet the inoculation method affects the establishment and function of this symbiosis remains unclear. Therefore, we conducted an experiment examining the influence of different AMF (Funneliformis mosseae (F. mosseae) and Claroideoglomus etunicatum (C. etunicatum) their combination (Mixed)) on C. migao seedlings. AMF colonization rates, root vigor, seedling growth and biomass, soil physicochemical properties, and enzyme activities were measured. The results showed that all three AMF treatments significantly enhanced the growth, plant biomass, and soil enzyme activity of C. migao seedlings. Among them, G. etunicatum demonstrated the most effective overall promotion. Obviously, the application of AMF, particularly G. etunicatum, can enhance the drought resistance of C. migao, which supports its large-scale cultivation and offers insights for ecological restoration in semi-arid regions.”

Question 2: The Introduction provides adequate background but includes methodological details that should be moved to the Materials and Methods section. It should be summarized and clearly state the research objectives.

Response: Thank you for your professional advice, we have moved the methodological details to the Materials and Methods section and re-written the Introduction with your recommendations. The updated version is provided below.

“Global warming and rapid industrial development are increasing the frequency of extreme weather events, which subject plants in terrestrial ecosystems to abiotic stresses—including extreme drought, high temperature, and flooding—that disrupt key physiological processes such as photosynthesis, osmoregulation, and membrane lipid peroxidation. Although water shortage impairs plant growth and productivity via multiple pathways, the limited self-protective capacity of plants is compensated by rhizosphere microorganisms, which play a key role in plant adaptation to stressful environments. Due to their sensitivity to climatic and edaphic factors, these microorganisms also reflect ecosystem stability and function. Consequently, investigating the rhizosphere interface—where plant and microbial responses converge—is imperative for advancing predictive frameworks of plant performance under environmental change.

Arbuscular mycorrhizal fungi (AMF), an ancient endophytic fungal group, exist widely in nature and can form symbiotic relationships with more than 90% of plants in terrestrial ecosystems. Previous studies have demonstrated that AMF can establish a bidirectional reward relationship with plants, AMF form a symbiotic relationship with host plants in which the plant allocates 15–20% of its photosynthetic products to support fungal growth, thereby enriching the soil microbial environment. In return, the extensive hyphal network of AMF expands the root absorption area, enhancing nutrient uptake, improving soil structure, increasing photosynthetic efficiency, and ultimately promoting plant biomass accumulation. This discovery provides new ideas for the enhancement of plant stress resistance in harsh environments. Accumulating evidence indicates that AMF enhance water and mineral uptake in terrestrial plants by modifying root morphology and the rhizosphere environment. Additionally, they alleviate drought stress through multiple pathways, including the regulation of drought-responsive genes, improved photosynthetic efficiency, and enhanced antioxidant and osmoregulatory capacity in host plants.

Cinnamomum migao (C. migao) is a tree species used for vegetation restoration and medicinal resource: its fruit extracts, containing diverse active components, are applied clinically for cardiovascular, neoplastic, and inflammatory disorders. Unfortunately, in recent years, due to excessive and unregulated logging and impaired natural regeneration, the wild resource reserve of C.migao has declined rapidly. Its slow natural regeneration cannot meet the rising market demand, resulting in resource shortages that must be addressed through large-scale artificial propagation. Therefore, how to improve the growth of C. migao under drought stress is a critical issue that urgently needs to be addressed.”

Question 3: The Materials and Methods section is generally clear but requires minor clarification. Define all abbreviations, specify sample sizes, and indicate measurement units for all parameters.

Response: Considering your suggestion, we have defined all abbreviations, sample sizes, and indicate measurement units for all parameters in Materials and Methods section and under each figure.

For example:

“Cinnamomum migao H.W.Li (C. migao), a tree in the Cinnamomum genus of Lauraceae family, is a medicinally important tree species endemic to southwest China. Arbuscular mycorrhizal fungi (AMF) symbiosis mitigates drought stress in plants, yet the inoculation method affects the establishment and function of this symbiosis remains unclear. Therefore, we conducted an experiment examining the influence of different AMF (Funneliformis mosseae (F. mosseae) and Claroideoglomus etunicatum (C. etunicatum) their combination (Mixed)) on C. migao seedlings.”

“In the first stage, all seedlings were well watered for 120 days (prior stress, PS) to ensure adequate colonization of AMF. Then they were subjected to drought stress (SS) for the next 30 days (seedlings began wilting), and then rewatered (REC) for the next 30 days (seedlings resumed growth).”

Question 4: The Results are informative but need better organization under clear subheadings and with clearer figures. Define symbols, add units for all parameters, and ensure figure consistency.

Response: We are very sorry for our negligence of definitions for all symbols and markings within the figures. They have now been added in compliance with your suggestions below.

“Fig.1 AMF colonization and root vigor of C. migao seedlings at different growth stages. CK, no inoculation; C. etunicatum, seedlings inoculated with C. etunicatum; F. mosseae, seedlings inoculated with F. mosseae; and Mixed, seedlings inoculated with C. etunicatum and F. mosseae. PS: prior stress, SS: subjected to drought stress; REC: rewatered; DS: drought stress. Different letters (a, b, c, d) indicate a significant difference by Tukey’s post hoc test. ns, not significant; * P < 0.05 and ** P< 0.01. Values are expressed as the SE (n = 5, which are treatment replicates).”

Question 5: The Discussion includes many long and repetitive sentences. Use different wording and condense overlapping ideas. Including an outlook on future research would strengthen this section.

Response: Considering your suggestion, the Discussion has been appropriately streamlined and revised. Furthermore, a section outlining prospects for future research has been added to the conclusion.

“We evaluated the effects of F. mosseae, C. etunicatum, and their combined inoculation on the growth, soil physicochemical properties, and enzyme activities of C. migao seedlings under both well-watered and drought-stress conditions. The results showed that selected fungi colonizing the plant rhizosphere can form symbioses with C. migao seedlings and AMF facilitated plant growth and soil nutrient acquisition. C. etunicatum was the more effective species for improving plant growth and nutrient uptake among all treatments, regardless of the water regime. In the future, we should conduct more research on the biological characteristics of each type of AMF to understand its ecological response and impact under drought stress. And further research on the changes in plant hormones and signal transduction substances mediated by AMF.”

Once again, thank you very much for your constructive comments and suggestions, which have been invaluable in helping us improve both the language clarity and academic depth of the manuscript. Your insights have significantly strengthened the quality of our paper.

Yours sincerely,

Gang Hao

Suzhou Vocational University

Wuzhong District, Suzhou 215100, China

Tel: 86+15370078353

E-mail: 626052286@qq.com

---

## [Decision Letter · Decision Letter 1]

6 Apr 2026

Arbuscular mycorrhizal fungi improve drought toleration in Cinnamomum migao H.W.Li seedlings by increasing plant growth, nutrient uptake and biomass accumulation

PONE-D-25-45963R1

Dear Dr. Hao,

We’re pleased to inform you that your manuscript has been judged scientifically suitable for publication and will be formally accepted for publication once it meets all outstanding technical requirements.

Kind regards,

Debasis Mitra

Academic Editor

PLOS One

Additional Editor Comments (optional):

Reviewers' comments:

Reviewer #1: (No Response)

Reviewer #2: Thank you for the revised version of the manuscript. The authors have addressed the previous comments satisfactorily and the manuscript has improved accordingly. I have no further comments.

---

## [Editor Report · Acceptance letter]

PONE-D-25-45963R1

PLOS One

Dear Dr. Gang,

I'm pleased to inform you that your manuscript has been deemed suitable for publication in PLOS One. Congratulations! Your manuscript is now being handed over to our production team.

Kind regards,

on behalf of

Dr. Debasis Mitra

Academic Editor

PLOS One